# Creating Training Corpora for NLG Micro-Planning

## Abstract

In this paper, we focus on how to create data-to-text corpora which can support the learning of wide-coverage micro-planners i.e., generation systems that handle lexicalisation, aggregation, surface realisation, sentence segmentation and referring expression generation. We start by reviewing common practice in designing training benchmarks for Natural Language Generation. We then present a novel framework for semi-automatically creating linguistically challenging NLG corpora from existing Knowledge Bases. We apply our framework to DBpedia data and compare the resulting dataset with (Wen et al., 2016)'s dataset. We show that while (Wen et al., 2016)'s dataset is more than twice larger than ours, it is less diverse both in terms of input and in terms of text. We thus propose our corpus generation framework as a novel method for creating challenging data sets from which NLG models can be learned which are capable of generating text from KB data.

## 1 Introduction

To train Natural Language Generation (NLG) systems, various input-text corpora have been developed which associate (numerical, formal, linguistic) input with text. As discussed in detail in Section 2, these corpora can be classified into three main types namely, (i) domain specific corpora, (ii) benchmarks constructed from "Expert" Linguistic Annotations and (iii) crowdsourced benchmarks[1].

In this paper, we focus on how to create data-to-text corpora which can support the learning of wide-coverage micro-planners i.e., generation systems that handles such NLG subtasks as lexicalisation (mapping data to words), aggregation (exploiting linguistic constructs such as ellipsis and coordination to avoid repetition), surface realisation (using the appropriate syntactic constructs to build sentences), sentence segmentation and referring expression generation.

We start by reviewing the main existing types of NLG benchmarks and we argue for a crowdsourcing approach where data units are automatically built from an existing knowledge base and where text is crowdsourced from the data (Section 2).

We then propose a generic framework for semi-automatically creating training corpora for NLG (Section 3) from existing Knowledge Bases. In Section 4, we apply this framework to DBpedia data and we compare the resulting dataset with (Wen et al., 2016)'s using various metrics to evaluate the linguistic and computational adequacy of both datasets. By applying these metrics, we show that while (Wen et al., 2016)'s dataset is more than twice larger than ours, it is less diverse both in terms of input and in terms of text. We also compare the performance of a sequence-to-sequence model (Vinyals et al., 2015) on both datasets to estimate the complexity of the learning task induced by each dataset. We show that the performance of this neural model is much lower on the new data set than on the existing ones. We thus propose our corpus generation framework as a novel method for creating challenging data sets from which NLG

---

[1] We ignore here (Lebret et al., 2016)'s dataset which was created fully automatically from Wikipedia by associating infoboxes with text because this dataset fails to ensure an adequate match between data and text. We manually examined 50 input/output pairs randomly extracted from this dataset and did not find a single example where data and text matched. As such, this dataset is ill-suited for training micro-planners. Moreover, since its texts contain both missing and additional information, it cannot be used to train joint models for content selection and micro-planning either.

models can be learned which are capable of generating complex texts from KB data.

## 2 NLG Benchmarks

**Domain Specific Benchmarks.** Several domain specific data-text corpora have been built by researchers to train and evaluate NLG systems. In the sports domain, Chen and Mooney (2008) constructed a dataset mapping soccer games events to text which consists of 1,539 data-text pairs and a vocabulary of 214 words. For weather forecast generation, (Liang et al., 2009)'s dataset includes 29,528 data-text pairs with a vocabulary of 345 words. For the air travel domain, Ratnaparkhi (2000) created a dataset consisting of 5,426 data-text pairs with a richer vocabulary (927 words) and in the biology domain, the KBGen shared task (Banik et al., 2013) made available 284 data-text pairs where the data was extracted from an existing knowledge base and the text was authored by biology experts.

An important limitation of these datasets is that, because they are domain specific, systems learned from them are restricted to generating domain specific, often strongly stereotyped text (e.g., weather forecast or soccer game commentator reports). Arguably, training corpora for NLG should support the learning of wide-coverage generators. By nature however, domain specific corpora restrict the lexical and often the syntactic coverage of the texts to be produced and thereby indirectly limit the expressivity of the generators trained on them.

**Benchmarks Constructed from "Expert" Linguistic Annotations.** NLG benchmarks have also been proposed where the input data is either derived from dependency parse trees (SR'11 task, (Belz et al., 2011)) or constructed through manual annotation (AMR Corpus (Banarescu et al., 2012)). Contrary to the domain-specific data sets just mentioned, these corpora have a wider coverage and are large enough for training systems that can generate linguistically sophisticated text.

One main drawback of these benchmarks however is that their construction required massive manual annotation of text with complex linguistic structures (parse trees for the SR task and Abstract Meaning Representation for the AMR corpus). Moreover because these structures are complex, the annotation must be done by experts. It cannot be delegated to the crowd. In short, the creation of such benchmark is costly both in terms of time and in terms of expertise.

Another drawback is that, because the input representation derived from a text is relatively close to its surface form[2], the NLG task is mostly restricted to surface realisation (mapping input to sentences). That is, these benchmarks give very limited support for learning models that can handle micro-planning NLG subtasks such as lexicalisation, aggregation, sentence segmentation and referring expression generation.

**Crowdsourced Benchmarks.** More recently, data-to-text benchmarks have also been created by associating data units with text using crowdsourcing.

Wen et al. (2016) first created data by enumerating all possible combinations of 14 dialog act types (e.g., *request, inform*) and attribute-value pairs present in four small-size, hand-written ontologies about TVs, laptops, restaurants and hotels. They then use crowdsourcing to associate each data unit with a text. The resulting dataset is both large and varied (4 domains) and was successfully exploited to train neural and imitation learning data-to-text generator (Wen et al., 2016; Lampouras and Vlachos, 2016). Similarly, Novikova and Rieser (2016) described a framework for collecting data-text pairs using automatic quality control measures and evaluating how the type of the input representations (text vs pictures) impacts the quality of crowdsourced text.

The crowdsourcing approach to creating input-text corpora has several advantages.

First, it is low cost in that the data is produced automatically and the text is authored by a crowdworker. This is in stark contrast with the previous approach where expert linguists are required to align text with data.

Second, because the text is crowd-sourced from the data (rather than the other way round), there is an adequate match between text and data both semantically (the text expresses the information contained in the data) and computationally (the data is sufficiently different from the text to require the learning of complex generation operations such as sentence segmentation, aggregation and referring expression generation).

---

[2]For instance, the input structures made available by the shallow track of the SR task contain all the lemmas present in the corresponding text. In this case, the generation task is limited to determining (i) the linear ordering and (ii) the full form of the word in the input.

Third, by exploiting small hand-written ontologies to quickly construct meaningful artificial data, the crowdsourcing approach allows for the easy creation of a large dataset with data units of various size and bearing on different domains. This, in turn, allows for better linguistic coverage and for NLG tasks of various complexity since typically, inputs of larger size increases the need for complex microplanning operations.

## 3  A Framework for Creating Data-to-Text, Micro-Planning Benchmarks

While as just noted, the crowdsourcing approach presented in (Wen et al., 2016) has several advantages, it also has a number of shortcomings.

One important drawback is that it builds on artificial rather than "real" data i.e., data that would be extracted from an existing knowledge base. As a result, the training corpora built using this method cannot be used to train KB verbalisers i.e., generation systems that can verbalise KB fragments.

Another limitation concerns the shape of the input data. (Wen et al., 2016)'s data can be viewed as trees of depth one (a set of attributes-value pairs describing a single entity e.g., a restaurant or a laptop). As illustrated in Figure 1 however, there is a strong correlation between the shape of the input and the syntactic structure of the corresponding sentence. The path structure $T_1$ where B is shared by two predicates (*mission* and *operator*) will favour the use of a participial or a passive subject relative clause. In contrast, the branching structure $T_2$ will favour the use of a new clause with a pronominal subject or a coordinated VP. More generally, allowing for trees of deeper depth is necessary to indirectly promote the introduction in the benchmark of a more varied set of syntactic constructs to be learned by generators.

To address these issues, we introduce a novel method for creating data-to-text corpora from large knowledge bases such as DBPedia. Our method combines (i) a content selection module designed to extract varied, relevant and coherent data units from DBPedia with (ii) a crowdsourcing process for associating data units with human authored texts that correctly capture their meaning. Example 1 shows a data/text unit created by our method using DBPedia as input KB.

(1)  a.  *(John_E_Blaha birthDate 1942_08_26)*
         *(John_E_Blaha birthPlace San_Antonio)*
         *(John_E_Blaha occupation Fighter_pilot)*

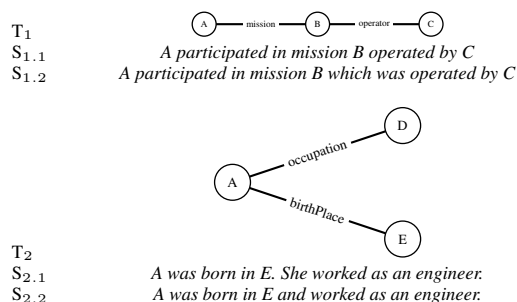

Figure 1: Input Shape and Linguistic Structures (A = Susan Helms, B = STS 78, C = NASA, D = engineer, E = Charlotte, North Carolina).

   b.  *John E Blaha, born in San Antonio on 1942-08-26, worked as a fighter pilot*

Our method has the following features.

First, it can be used to create a data-to-text corpus from any knowledge base where entities are categorised and there is a large number of entities belonging to the same category. As noted above, this means that the resulting corpus can be used to train KB verbalisers i.e., generators that are able to verbalise fragments of existing knowledge bases. It could be used for instance, to verbalise fragments of e.g., MusicBrainz[3], FOAF[4] or Linked-GeoData[5].

Second, as crowdworkers are required to enter text that matches the data and a majority vote validation process is used to eliminate mis-matched pairs, there is a direct match between text and data. This allows for a clear focus on the non content selection part of generation known as microplanning.

Third, because data of increasing size is matched with texts ranging from simple clauses to short texts consisting of several sentences, the resulting benchmark is appropriate for exercising the main subtasks of microplanning. For instance, in Example (1) above, given the input shown in (1a), generating (1b) involves lexicalising the *occupation* property as the phrase *worked as* (lexicalisation); using PP coordination (*born in San Antonio on 1942-08-26*) to avoid repeating the word *born* (aggregation); and verbalising the three triples using a single complex sentence including an apposition, a PP coordination and a transitive verb construction (sentence segmentation and surface realisation).

---

[3] https://musicbrainz.org/
[4] http://www.foaf-project.org/
[5] http://linkedgeodata.org/

### 3.1 DBPedia

To illustrate the functioning of our benchmark creation framework, we apply it to DBPedia. DBPedia is a multilingual knowledge base that was built from various kinds of structured information contained in Wikipedia (Mendes et al., 2012). This data is stored as RDF (Resource Description Format) triples of the form *(subject, property, object)* where the subject is a URI (Uniform Resource Identifier), the property is a binary relation and the object is either a URI or a literal value such as a string, a date or a number. We use an English version of the DBPedia knowledge base which encompasses 6.2M entities, 739 classes, 1,099 properties with reference values and 1,596 properties with typed literal values.[6]

### 3.2 Selecting Content

To create data units, we follow the procedure outlined in (Perez-Beltrachini et al., 2016) and sketched in Figure 2. This method can be summarised as follows.

First, DBPedia *category graphs* are extracted from DBPedia by retrieving up to 500 *entity graphs* for entities of the same category[7]. For example, we build a category graph for the Astronaut category by collecting, graphs of depth five for 500 entities of types astronaut.

Next, category graphs are used to learn bigram models of DBPedia properties which specify the probability of two properties co-occuring together. Three types of bi-gram models are extracted from category graphs using the SRILM toolkit: one model (*S*-Model) for bigrams occurring in sibling triples (triples with a shared subject); one model (*C*-Model) for bigrams occurring in chained triples (the object of one triple is the subject of the other); and one model (*M*-Model) which is a linear interpolation of the sibling and the chain model. The intuition is that these sibling and chain models capture different types of coherence, namely, topic-based coherence for the *S*-Model and discourse-based coherence for the *C*-Model.

Finally, the content selection task is formulated as an Integer Linear Programming (ILP) problem to select, for a given entity of category $C$ and its

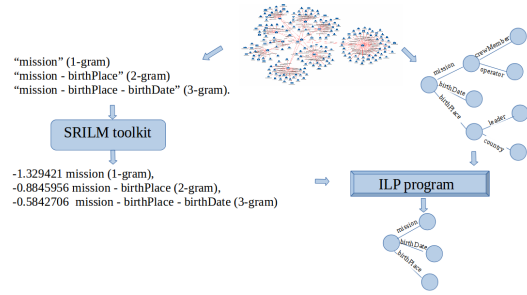

Figure 2: Extracting Data Units from DBPedia

entity graph $G_e$, subtrees of $G_e$ with maximal bigram probability and varying size (between 1 and 7 RDF triples).

We applied this content selection procedure to the DBPedia categories Astronaut (A), Building (B), Monument (M), University (U), Sports team (S) and Written work (W), using the three bi-gram models (*S*-Model, *C*-Model, *M*-Model) and making the number of triples required by the ILP constraint to occur in the output solutions vary between 1 and 7. The results are shown in Table 1. An input is a set of triples produced by the content selection module. The number of input is thus the number of distinct sets of triples produced by this module. In contrast, input patterns are inputs where subject and object have been abstracted over. That is, the number of input patterns is the number of distinct sets of properties present in the set of inputs. The number of properties is the number of distinct RDF properties occurring in the dataset. Similarly, the number of entities is the number of distinct RDF subjects and objects occurring in each given dataset.

| Category | A | B | M | U | S | W |
|---|---|---|---|---|---|---|
| #Inputs | 663 | 1220 | 333 | 508 | 1137 | 1207 |
| #I. Patterns | 546 | 369 | 300 | 432 | 184 | 277 |
| #Properties | 38 | 46 | 30 | 41 | 32 | 50 |
| #Entities | 74 | 278 | 47 | 75 | 264 | 224 |

Table 1: Data Statistics from content selection (A:Astronaut, B:Building, M:Monument, U:University, W:Written work, S:Sports team )

### 3.3 Associating Content with Text

We associate data with text using the Crowdflower platform[8]. We do this in four main steps as follows.

**1. Clarifying Properties.** One difficulty when collecting texts verbalising sets of DBPedia triples

---

[6]http://wiki.dbpedia.org/dbpedia-dataset-version-2015-10

[7]An entity graph for some entity $e$ is a graph obtained by traversing the DBPedia graph starting in $e$ and stopping at depth five.

[8]http://www.crowdflower.com

is that the meaning of DBPedia properties may be unclear. We therefore first manually clarified for each category being worked on, those properties which have no obvious lexicalisations (e.g., *crew1up* was replaced by *commander*).

**2. Getting Verbalisations for Single Triples.** Next, we collected three verbalisations for data units of size one, i.e. single triples consisting of a subject, a property and an object. For each such input, crowdworkers were asked to produce a sentence verbalising its content. We used both *a priori* automatic checks to prevent spamming and *a posteriori* manual checks to remove incorrect verbalisations. We also monitored crowdworkers as they entered their input and banned those who tried to circumvent our instructions and validators. The automatic checks comprise 12 custom javascript validators implemented in the CrowdFlower platform to block contributor answers which fail to meet requirements such as the minimal time a contributor should stay on page, the minimal length of the text produced, the minimal match of tokens between a triple and its verbalisation and various format restrictions used to detect invalid input. The exact match between a triple and its verbalisation was also prohibited. In addition, after data collection was completed, we manually checked each data-text pair and eliminated from the data set any pair where the text either did not match the information conveyed by the triple or was not a well-formed English sentence.

**3. Getting Verbalisations for Input containing more than one Triple.** The verbalisations collected for single triples were used to construct input with bigger size. Thus, for input with a number of triples more than one, the crowd was asked to merge the sentences corresponding to each triple (obtained in step 2) into a natural sounding text. In such a way, we diminish the risk of having misinterpretations of the original semantics of a data unit. Contributors were also encouraged to change the order, and the wording of sentences, while writing their texts. For each data unit, we collected three verbalisations.

**4. Verifying the Quality of the Collected Texts.** The verbalisations obtained in Step 3 were verified through crowdsourcing. Each verbalisation collected in Step 3 was displayed to CrowdFlower contributors together with the corresponding set of triples. Then the crowd was asked to assess its

fluency, semantic adequacy, and grammaticality. Those criteria were checked by asking the following three questions: *Does the text sound fluent and natural?*, *Does the text contain all and only the information from the data?*, *Is the text good English (no spelling or grammatical mistakes)?*. We collected five answers per verbalisation. A verbalisation was considered as bad, if it received three negative answers in at least one criterion. After the verification step, the total corpus loss was of 8.7%.

Table 2 shows some statistics about the text obtained using our crowdsourcing procedure.

# 4 Comparing Benchmarks

We now compare a dataset created using our dataset creation framework (henceforth DBPNLG) with (Wen et al., 2016)'s dataset[9] (henceforth, RNNLG). Example 2 shows a sample data-text pair taken from the RNNLG dataset. The DBPNLG dataset has been uploaded with this submission.

(2) Dialog Moves
recommend(name=caerus 33;type=television; screensizerange=medium;family=t5;hasusbport=true)
*The caerus 33 is a medium television in the T5 family that's USB-enabled*

As should be clear from the discussion in Section 2 and 3, both datasets are similar in that, in both cases, data is built from ontological information and text is crowdsourced from the data. An important difference between the two datasets is that, while the RNNLG data was constructed by enumerating possible combinations of dialog act types and attribute-value pairs, the DBPNLG data is created using a sophisticated content selection procedure geared at producing sets of data units that are relevant for a given ontological category and that are varied in terms of size, shape and content (Perez-Beltrachini et al., 2016). We now investigate the impact of this difference on the two datasets (DBPNLG and RNNLG). To assess the degree to which both datasets support the generation of linguistically varied text requiring complex microplanning operations, we examine a number of data and text related metrics. We also compare the results of an out-of-the-box sequence-to-sequence model as a way to estimate the complexity of the learning task induced by each dataset.

## 4.1 Data Comparison

*Terminology.* The attributes in (Wen et al., 2016)'s dataset can be viewed as binary relations between

---

[9]https://github.com/shawnwun/RNNLG

| # Triples | 1 | 2 | 3 | 4 | 5 | 6 | 7 |
|---|---|---|---|---|---|---|---|
| # Tokens | 4/30/10.48 | 11/45/22.97 | 7/37/16.96 | 17/60/36.38 | 14/53/29.61 | 29/80/49.14 | 24/73/42.95 |
| # Sentences | 1/2/1.00 | 1/4/1.23 | 1/3/1.02 | 1/5/2.05 | 1/4/1.64 | 1/6/2.85 | 1/5/2.42 |

Table 2: Text Statistics from crowdsourcing (min/max/avg).

the object talked about (a restaurant, a laptop, a TV or a hotel) and a value. Similarly, in DBPNLG, DBpedia RDF properties relate a subject entity to an object which can be either an entity or a datatype value. In what follows, we refer to both as *attributes*.

Table 3 shows several statistics which indicate that, while the RNNLG dataset is larger than DBPNLG, DBPNLG is much more diverse in terms of attributes, input patterns and input shapes.

**Number of attributes.** As illustrated in Example (3) below, different attributes can be lexicalised using different parts of speech. A dataset with a larger number of attributes is therefore more likely to induce texts with greater syntactic variety.

(3) Verb: *X title Y / X served as Y*
Relational noun: *X nationality Y / X's nationality is Y*
Preposition: *X country Y / X is in Y*
Adjective: *X nationality USA / X is American*

As shown in Table 3, DBPNLG has a more diverse attribute set than RNNLG both in absolute (172 attributes in DBPNLG against 108 in RNNLG) and in relative terms (RNNLG is a little more than twice as large as DBPNLG).

**Number of Input Patterns.** Since attributes may give rise to lexicalisation with different parts of speech, the sets of attributes present in an input (input pattern[10]) indirectly determine the syntactic realisation of the corresponding text. Hence a higher number of input patterns will favour a higher number of syntactic realisations. This is exemplified in Example (4) where two inputs with the same number of attributes give rise to texts with different syntactic forms. While in Example (4a), the attribute set { *country, location, startDate* } is realised by a passive (*is located*), an apposition (*Australia*) and a deverbal nominal (*its construction*), in Example (4b), the attribute set { *almaMater, birthPlace, selection* } induced a passive (*was born*) and two VP coordinations (*graduated and joined*).

(4) a. (*'108_St_Georges_Terrace location Perth', 'Perth country Australia', '108_St_Georges_Terrace startDate 1981'*)
*country, location, startDate*
*108 St. Georges Terrace is located in Perth, Australia. Its construction began in 1981.*
*passive, apposition, deverbal nominal*

b. (*'William_Anders selection 1963', 'William_Anders birthPlace British_Hong_Kong', 'William_Anders almaMater "AFIT, M.S. 1962"'*)
*almaMater, birthPlace, selection*
*William Anders was born in British Hong Kong, graduated from AFIT in 1962, and joined NASA in 1963.*
*passive, VP coordination, VP coordination*

Again, despite the much larger size of the RNNLG dataset, the number of input patterns in both datasets is almost the same. That is, the relative variety in input patterns is higher in DBPNLG.

**Number of Input / Number of Input Patterns.** The ratio between number of inputs and the number of input patterns has an important impact both in terms of linguistic diversity and in terms of learning complexity. A large ratio indicates a "repetitive dataset" where the same pattern is instantiated a high number of times. While this facilitates learning, this also reduces linguistic coverage (less combinations of structures can be learned) and may induce overfitting. Note that because datasets are typically delexicalised when training NLG models (cf. e.g., (Wen et al., 2015; Lampouras and Vlachos, 2016)) , at training time, different instantiations of the same input pattern reduce to identical input.

The two datasets markedly differ on this ratio which is five times lower in DBPNLG. While in DBPNLG, the same pattern is instantiated in average 2.40 times, it is instantiated 10.31 times in average in RNNLG. From a learning perspective, this means that the RNNLG dataset facilitates learning but also makes it harder to assess how well systems trained on it can generalise to handle unseen input.

**Input Shape.** As mentioned in Section 3, in the RNNLG dataset, all inputs can be viewed as trees of depth one while in the DBPNLG dataset, input may have various shapes. As a result, RNNLG texts will be restricted to syntactic forms which permit expressing such multiple predications of the same

---

[10]Recall from section 3 that input patterns are inputs where subjects and objects have been remove thus, in essence, an input pattern is the set of all the attributes occurring in a given input.

entity e.g., subject relative clause, VP and sentence coordination etc. In contrast, the trees extracted by the DBPNLG content selection procedure may be of depth five and therefore allow for further syntactic constructs such as object relative clause and passive participials (cf. Figure 1).

We can show this empirically as well that DBPNLG is far more diverse than RNNLG in terms of input shapes. The RNNLG dataset has only 6 distinct shapes and all of them are of depth 1, i.e., all (attribute, value) pairs in an input are siblings to each other. In contrast, the DBPNLG dataset has 58 distinct shapes, out of which only 7 shapes are with depth 1, all others have depth more than 1 and they cover 49.6% of all inputs.

| | DbpNLG | RNNLG |
|---|---|---|
| Nb. Input | 5068 | 22225 |
| Nb. Data-Text Pairs | 13339 | 30842 |
| Nb. Domains | 6 | 4 |
| Nb. Attributes | 172 | 108 |
| Nb. Input Patterns | 2108 | 2155 |
| Nb. Input / Nb Input Pattern | 2.40 | 10.31 |
| Nb. Input Shapes | 58 | 6 |

Table 3: Comparing DBPNLG and RNN. Attributes are properties in RDF triples or slots in dialog acts.

## 4.2 Text Comparison

Table 4 gives some statistics about the texts contained in each dataset.

(5) a. *(Alan_Bean birthDate "1932-03-15")*
 *Alan Bean was born on March 15, 1932*

(6) a. *('Alan_Bean nationality United_States', 'Alan_Bean birthDate "1932-03-15"', 'Alan_Bean almaMater "UT Austin, B.S. 1955"', 'Alan_Bean birthPlace Wheeler,_Texas', 'Alan_Bean selection 1963')*
 *Alan Bean was an American astronaut, born on March 15, 1932 in Wheeler, Texas. He received a Bachelor of Science degree at the University of Texas at Austin in 1955 and was chosen by NASA in 1963.*

As illustrated by the contrast between example 5 and 6 above, text length (number of tokens per text) and the number of sentences per text are strong indicators of the complexity of the generation task. We use the Stanford Part-Of-Speech Tagger and Parser version 3.5.2 (date 2015-04-20) to tokenize and to perform sentence segmentation on text. As shown in Table 4, DBPNLG's texts are longer both in terms of tokens and in terms of number of sentences per text. Another difference between the two datasets is that DBPNLG contains a

higher number of text per input thereby providing a better basis for learning paraphrases.

| | DbpNLG | RNNLG |
|---|---|---|
| Nb. Text / Input | 2.63 | 1.38 |
| Text Length (avg/median/min/max) | 24.36/23/4/80 | 18.37/19/1/76 |
| Nb. Sentence / Text (avg/median/min/max) | 1.45/1/1/6 | 1.25/1/1/6 |
| Nb. Tokens | 290479 | 531871 |
| Nb. Types | 2992 | 3524 |
| Lexical Sophistication | 0.69 | 0.54 |
| CTTR | 3.93 | 3.42 |

Table 4: Text Statistics from DBPNLG and RNNLG.

The size and the content of the vocabulary is another important factor in ensuring the learning of wide coverage generators. While a large vocabulary makes the learning problem harder, it also allows for larger coverage. DBPNLG exhibits a higher corrected type-token ratio (CTTR), which indicates greater lexical variety, and higher lexical sophistication (LS). Lexical sophistication measures the proportion of relatively unusual or advanced word types in the text. In practice, LS is the proportion of lexical word types (lemma) which are not in the list of 2,000 most frequent words generated from the British National Corpus[11]. Type-token ratio (TTR) is a measure of diversity defined as the ratio of the number of word types to the number of words in a text. To address the fact that this ratio tends to decrease with the size of the corpus, corrected TTR can be used to control for corpus size. It is defined as $T/\sqrt{2N}$, where $T$ is the number of types and $N$ the number of tokens.

Overall, the results shown in Table 4 indicate that DBPNLG texts are both lexically more diverse (higher corrected type/token ratio) and more sophisticated (higher proportion of unfrequent words) than RNNLG's. They also show a proportionately larger vocabulary for DBPNLG (2992 types for 290479 tokens in DBPNLG against 3524 types for 531871 tokens in RNNLG).

## 4.3 Neural Generation

Richer and more varied datasets are harder to learn from. As a proof-of-concept study of the comparative difficulty of the two datasets with respect to machine learning, we compare the performance of a sequence-to-sequence model for generation on both datasets. In particular, we use a multi-layered sequence-to-sequence model with an at-

---

[11]We compute LS and CTTR using the Lexical Complexity Analyzer developed by Lu (2012).

tention mechanism (Vinyals et al., 2015).[12] The model was trained with 3 layers of 512 units each. To allow for a fair comparison, we use a similar amount of data (13K data-text pairs) for both datasets. As RNNLG is bigger in size than DBPNLG, we constructed a balanced sample of RNNLG which included equal number of instances per category (*tv, laptop*, etc). We use a 3:1:1 ratio for training, developement and testing. The training was done in two delexicalisation modes: *fully* and *name only*. In case of fully delexicalisation, all entities were replaced by their generic terms, whereas in *name only* mode only subjects were modified in that way. For instance, the triple *FC Köln manager Peter Stöger* was delexicalised as *SportsTeam manager Manager* in the first mode, and as *SportsTeam manager Peter Stöger* in the second mode. The delexicalisation in sentences was done using the exact match between entities and tokens.

Table 5 shows the perplexity results. In both modes, RNNLG yielded lower scores than DBPNLG. This is inline with the observations made above concerning the higher data diversity, larger vocabulary and more complex texts of DBPNLG. Similary, the BLEU score of the generated sentences (Papineni et al., 2002) is lower for DBPNLG suggesting again a dataset that is more complex and therefore more difficult to learn from.

|  | Delexicalisation Mode | DbpNLG | RNNLG |
|---|---|---|---|
| Perplexity | Fully | 27.41 | 17.42 |
|  | Name only | 25.39 | 23.93 |
| BLEU | Fully | 0.19 | 0.26 |
|  | Name only | 0.10 | 0.27 |

Table 5: Perplexity and BLEU scores.

## 5 Conclusion

We presented a framework for building NLG data-to-text training corpora from existing knowledge bases.

One feature of our framework is that datasets created using this framework can be used for training and testing KB verbalisers an in particular,

---

[12]We used the TensorFlow code available at https://github.com/tensorflow/models/tree/master/tutorials/rnn/translate. Alternatively, we could have used (Wen et al., 2016)'s implementation which is optimised for generation. However the code is geared toward dialog acts and modifying it to handle RDF triples is non trivial. Since the comparison aims at examining the relative performance of the same neural network on the two datasets, we used the tensor flow implementation instead.

verbalisers for RDF knowledge bases. Following the development of the semantic web, many large scale datasets are encoded in the RDF language (e.g., MusicBrainz, FOAF, LinkedGeoData) and official institutions[13] increasingly publish their data in this format. In this context, our framework is useful both for creating training data from RDF KB verbalisers and to increase the number of datasets available for training and testing NLG.

Another important feature of our framework is that it permits creating semantically and linguistically diverse datasets which should support the learning of lexically and syntactically, wide coverage micro-planners. We applied our framework to DBpedia data and showed that although twice smaller than the largest corpora currently available for training data-to-text microplanners, the resulting dataset is more semantically and linguistically diverse. Despite the disparity in size, the number of attributes is comparable in the two datasets. The ratio between input and input patterns is five times lower in our dataset thereby making learning harder but also diminishing the risk of overfitting and providing for wider linguistic coverage. Conversely, the ratio of text per input is twice higher thereby providing better support for learning paraphrases.

We are currently working on further extending the DBPNLG dataset and once completed, will make it available as part of a shared task for evaluating data-to-text micro-planners. While we only report on a dataset developed using 6 DBpedia categories, we have collected content for 14 further categories using the content selection procedure described in Section 3 and will collect the corresponding texts using our selective crowdsourcing procedure.

Recently, several sequence-to-sequence models have been proposed for generation. Our experiments suggest that these are not optimal when it comes to generate linguistically complex texts from rich data. More generally, they indicate that the data-to-text corpora built by our framework are challenging for such models. We hope that the DBPNLG dataset which we will make available in the shared task will drive the deep learning community to take up this new challenge and work on the development of neural generators that can handle the generation of linguistically rich texts.

---

[13]See http://museum-api.pbworks.com for examples.

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
