# Peer review of "Creating Training Corpora for NLG Micro-Planners"

_ACL 2017 — decision unknown_

[Official Review · Reviewer 1 · rating 4 · confidence 4]
soundness 5 · originality 5 · clarity 4 · impact 3 · substance 2 · appropriateness 4 · meaningful comparison 3 · presentation format Poster

- Strengths:

This paper presents a step in the direction of developing more challenging
corpora for training sentence planners in data-to-text NLG, which is an
important and timely direction. 

- Weaknesses:

It is unclear whether the work reported in this paper represents a substantial
advance over Perez-Beltrachini et al.'s (2016) method for selecting content. 
The authors do not directly compare the present paper to that one. It appears
that the main novelty of this paper is the additional analysis, which is
however rather superficial.

It is good that the authors report a comparison of how an NNLG baseline fares
on this corpus in comparison to that of Wen et al. (2016).  However, the
BLEU scores in Wen et al.'s paper appear to be much much higher, suggesting
that this NNLG baseline is not sufficient for an informative comparison.

- General Discussion:

The authors need to more clearly articulate why this paper should count as a
substantial advance over what has been published already by Perez-Beltrachini
et al, and why the NNLG baseline should be taken seriously.  In contrast to
LREC, it is not so common for ACL to publish a main session paper on a corpus
development methodology in the absence of some new results of a system making
use of the corpus.

The paper would also be stronger if it included an analysis of the syntactic
constructions in the two corpora, thereby more directly bolstering the case
that the new corpus is more complex.  The exact details of how the number of
different path shapes are determined should also be included, and ideally
associated with the syntactic constructions.

Finally, the authors should note the limitation that their method does nothing
to include richer discourse relations such as Contrast, Consequence,
Background, etc., which have long been central to NLG. In this respect, the
corpora described by Walker et al. JAIR-2007 and Isard LREC-2016 are more
interesting and should be discussed in comparison to the method here.

References

Marilyn Walker, Amanda Stent, François Mairesse, and
Rashmi Prasad. 2007. Individual and domain adaptation
in sentence planning for dialogue. Journal of
Artificial Intelligence Research (JAIR), 30:413–456.

Amy Isard, 2016. “The Methodius Corpus of Rhetorical Discourse
Structures and Generated Texts” , Proceedings of the Tenth Conference
on Language Resources and Evaluation (LREC 2016), Portorož, Slovenia,
May 2016.

---
Addendum following author response:

Thank you for the informative response.  As the response offers crucial
clarifications, I have raised my overall rating.  Re the comparison to
Perez-Beltrachini et al.: While this is perhaps more important to the PC than
to the eventual readers of the paper, it still seems to this reviewer that the
advance over this paper could've been made much clearer.  While it is true that
Perez-Beltrachini et al. "just" cover content selection, this is the key to how
this dataset differs from that of Wen et al.  There doesn't really seem to be
much to the "complete methodology" of constructing the data-to-text dataset
beyond obvious crowd-sourcing steps; to the extent these steps are innovative
or especially crucial, this should be highlighted.  Here it is interesting that
8.7% of the crowd-sourced texts were rejected during the verification step;
related to Reviewer 1's concerns, it would be interesting to see some examples
of what was rejected, and to what extent this indicates higher-quality texts
than those in Wen et al.'s dataset.  Beyond that, the main point is really that
collecting the crowd-sourced texts makes it possible to make the comparisons
with the Wen et al. corpus at both the data and text levels (which this
reviewer can see is crucial to the whole picture).

Re the NNLG baseline, the issue is that the relative difference between the
performance of this baseline on the two corpora could disappear if Wen et al.'s
substantially higher-scoring method were employed.  The assumption that this
relative difference would remain even with fancier methods should be made
explicit, e.g. by acknowledging the issue in a footnote.  Even with this
limitation, the comparison does still strike this reviewer as a useful
component of the overall comparison between the datasets.

Re whether a paper about dataset creation should be able to get into ACL
without system results:  though this indeed not unprecedented, the key issue is
perhaps how novel and important the dataset is likely to be, and here this
reviewer acknowledges the importance of the dataset in comparison to existing
ones (even if the key advance is in the already published content selection
work).

Finally, this reviewer concurs with Reviewer 1 about the need to clarify the
role of domain dependence and what it means to be "wide coverage" in the final
version of the paper, if accepted.

[Official Review · Reviewer 2 · rating 3 · confidence 4]
soundness 5 · originality 5 · clarity 4 · impact 3 · substance 3 · appropriateness 5 · meaningful comparison 3 · presentation format Oral Presentation

- Strengths:
* Potentially valuable resource
* Paper makes some good points

- Weaknesses:
* Awareness of related work (see below)
* Is what the authors are trying to do (domain-independent microplanning) even
possible (see below)
* Are the crowdsourced texts appropriate (see below)

- General Discussion:
This is an interesting paper which presents a potentially valuable resource,
and I in many ways I am sympathetic to it.  However, I have some high-level
concerns, which are not addressed in the paper.  Perhaps the authors can
address these in their response.

(1) I was a bit surprised by the constant reference and comparison to Wen 2016,
which is a fairly obscure paper I have not previously heard of.  It would be
better if the authors justified their work by comparison to well-known corpora,
such as the ones they list in Section 2. Also, there are many other NLG
projects that looked at microplanning issue when verbalising DBPedia, indeed
there was a workshop in 2016 with many papers on NLG and DBPedia
(https://webnlg2016.sciencesconf.org/  and
http://aclweb.org/anthology/W/W16/#3500); see also previous work by Duboue and
Kutlak.  I would like to see less of a fixation on Wen (2016), and more
awareness of other work on NLG and DBPedia.

(2) Microplanning tends to be very domain/genre dependent.  For example,
pronouns are used much more often in novels than in aircraft maintenance
manuals.   This is why so much work has focused on domain-dependent resources. 
  So there are some real questions about whether it is possible even in theory
to train a "wide-coverage microplanner".  The authors do not discuss this at
all; they need to show they are aware of this concern.

(3) I would be concerned about the quality of the texts obtained from
crowdsourcing.              A lot of people dont write very well, so it is not at all
clear
to me that gathering example texts from random crowdsourcers is going to
produce a good corpus for training microplanners.  Remember that the ultimate
goal of microplanning is to produce texts that are easy to *read*.  Imitating
human writers (which is what this paper does, along with most learning
approaches to microplanning) makes sense if we are confident that the human
writers have produced well-written easy-to-read texts.              Which is a
reasonable
assumption if the writers are professional journalists (for example), but a
very dubious one if the writers are random crowdsourcers.

From a presentational perspective, the authors should ensure that all text in
their paper meets the ACL font size criteria.  Some of the text in Fig 1 and
(especially) Fig 2 is tiny and very difficult to read; this text should be the
same font size as the text in the body of the paper.

I will initially rate this paper as borderline.  I look forward to seeing the
author's response, and will adjust my rating accordingly.